# Identification of Foulants on Polyethersulfone Membranes Used to Remove Colloids and Dissolved Matter from Paper Mill Treated Effluent

**Mayko Rannany S. Sousa** [1], **Jaime Lora-García** [1], **María-Fernanda López-Pérez** [1,*] **and Marc Heran** [2]

[1] Instituto de Seguridad Industrial, Radiofísica y Medioambiental (ISIRYM), Universitat Politècnica de València (UPV), Plaza Ferrándiz y Carbonell, s/n, 03801 Alcoy, Spain; maysanso@doctor.upv.es (M.R.S.S.); jlora@iqn.upv.es (J.L.-G.)

[2] IEM, Université Montpellier, CNRS, ENSCM, 2 rue Monteil, 34033 Montpellier, France; marc.heran@umontpellier.fr

[*] Correspondence: malope1@iqn.upv.es

**Abstract:** In this study, membrane fouling caused by paperboard mill treated effluent (PMTE) was investigated based on a dead-end ultrafiltration (UF) pilot-scale study. The membranes employed were commercial hydrophobic UF membranes made of polyethersulfone (PES) with a molecular weight cut-off of 10 kDa, 50 kDa, and 100 kDa. Membrane fouling mechanism during dead-end filtration, chemical analysis, field emission scanning electron microscopy (FESEM), energy-dispersive spectrophotometry (EDS), attenuated total reflection-Fourier transform infrared (ATR-FTIR) spectroscopy and 3D fluorescence excitation–emission matrix (3DEEM) analysis were applied to understand which fraction of the dissolved and colloidal substances (DCS) caused the membrane fouling. The results indicated that the phenomenon controlling fouling mechanism tended to be cake layer formation ($R^2 \geq 0.98$) for all membranes tested. The 3DEEM results indicate that the majority of the organic foulants with fluorescence characteristics on the membrane were colloidal proteins (protein-like substances I+II) and macromolecular proteins (soluble microbial products, SMP-like substances). In addition, polysaccharide (cellulosic species), fatty and resin acid substances were identified on the fouled membrane by the ATR–FTIR analysis and play an important role in membrane fouling. In addition, the FESEM and EDS analyses indicate that the presence of inorganic foulants on the membrane surfaces, such as metal ions and especially $Ca^{2+}$, can accelerate membrane fouling, whereas Mg and Si are linked to reversible fouling.

**Keywords:** ultrafiltration; paper mill effluent; membrane fouling; foulants identification

## 1. Introduction

The paper industry has an important place in the Spanish economy as Spain is one of the European leaders in paperboard recycling, with more than 84% of the raw materials used by the paper industry coming from recovered paper [1]. However, this manufacturing process consumes large amounts of freshwater and consequently generates a significant amount of wastewater with a high biodegradable organic matter loading [2,3].

Moreover, the diversity of raw materials used in recycled cardboard papers requires a great knowledge of water circuits to adopt an adequate decision on internal water recycling. In a conventional cardboard paper mill, several water loops can be present with different uses in the papermaking process (Figure 1). Short circuits are used for water recycling without treatment. Other internal water circulation, named long circuits, are carried out after flotation or clarification as the first wastewater

treatment process followed by other conventional wastewater treatments [4–6]. The most utilized of them used to be biological treatments, but in many cases, they are insufficient to comply with the most stringent environmental regulations on effluent discharge or to obtain a suitable quality for the process water to be reused in papermaking.

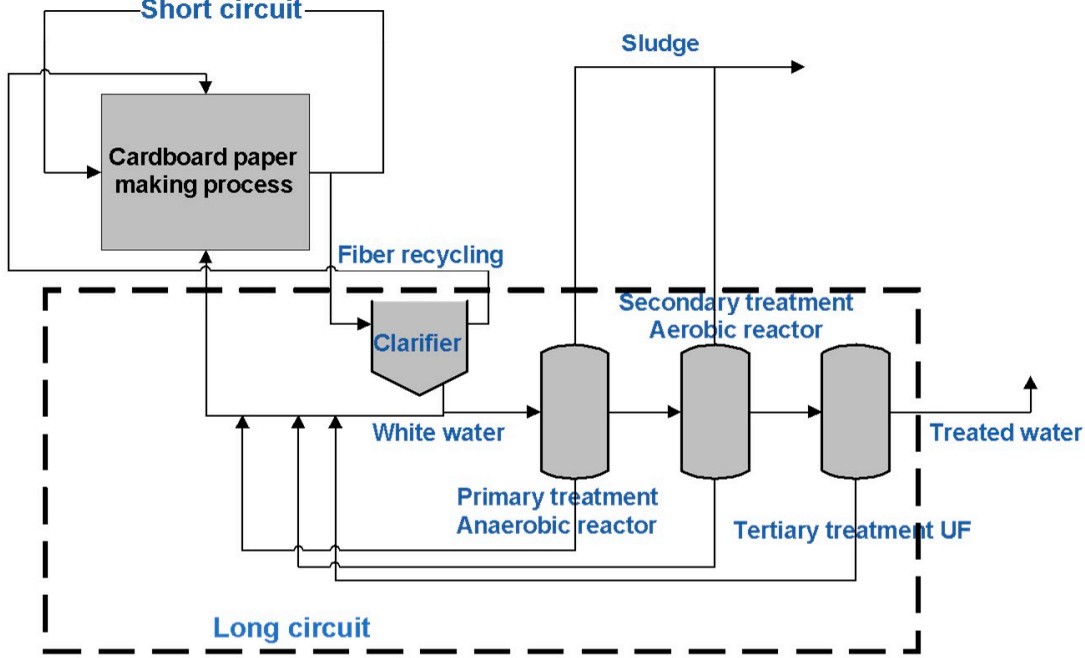

**Figure 1.** Flow chart of the recovery process of water in the cardboard papermaking process.

Due to these limitations in achieving adequate water quality, membrane separation technology has attracted increasing attention as a tertiary treatment for these treated effluents from paper mills, particularly as it facilitates subsequent effluent recycling [7–9]. However, membrane fouling continues to be one of the main limitations and challenges to the wider scale application of membrane technology in paper mill effluent treatment. Thus, to efficiently control membrane fouling in the ultrafiltration (UF) process, it is important to understand which components in the effluent play a major role in membrane fouling.

Many authors have studied ultrafiltration as the main treatment to remove the dissolved and colloidal substances (DCS) concentrated during the recycling of white waters not treated by using biological treatment. The results show that in pulp and paper mill effluent (white water), DCS are considered to be the major membrane foulants [10–18].

Nevertheless, the analysis of the membrane fouling process when UF filtration is used as a tertiary treatment after the biological process in a paper mill has not been widely studied. Therefore, this paper is focused on analyzing UF fouling after biological treatment because it can modify organic substances presents in white waters, and additionally, it can produce new organic substances from biological sludge. Hence, a good knowledge of the characteristics of the substances present in these waters could be useful to help minimize membrane fouling, aimed at obtaining high-quality water from UF filtration that may reduce the use of freshwater. Furthermore, in the literature, investigations about foulants' characteristics had received little attention in comparison with studies focused on membrane performance and water quality. In this work, studies of membrane fouling problem, material that is retained at the membrane surface or inside the membrane pores, and analysis of DCS from biological effluent is presented.

Moreover, these foulants include some fluorescent organic compounds (FOC), such as protein-like substances, humic-like substances, and fulvic-like substances. These FOCs can be differentiated with high sensitivity due to their respective fluorescence properties in the ultraviolet and visible range [19].

To understand DCS composition from paper mill treated biological effluent (PMTE), batch dead-end UF with different polyethersulfone (PES) membranes were used. Resistance-in-series and Hermia's model were used to analyze the predominant fouling mechanism developed on each membrane and how it affects the permeate flux. Experimental filtration time was similar to industrial procedures avoiding long filtration time with extreme fouling, which is not recommended in the industrial process.

Techniques such as chemical analysis, field scanning electron microscopy (FESEM), energy-dispersive spectrophotometry (EDS), attenuated total reflection-Fourier transform infrared (ATR-FTIR) spectroscopy, and 3D fluorescence excitation–emission matrix (3DEEM) analysis were applied to understand which fraction of the DCS caused the reversible and irreversible fouling [18,20–24].

This study could be helpful to provide a more detailed understanding of the chemical composition, possible origins of membrane foulants, and fouling mechanism during the UF process for the removal of DCS coming from a paper mill secondary effluent. The result can be used as a tool to control the membrane fouling, and to know whether the UF process is an adequate technology or on the other hand, whether high energy consumption, operation and investment costs, caused by the fouling problems, restrict the use of this filtration type.

## 2. Materials and Methods

### 2.1. Feed Water

The feed water for the UF membrane experiments was obtained from a secondary clarifier from a wastewater treatment plant (WWTP) in a paper mill located in the south of the Valencian autonomous region in Spain (Figure 1). In this papermaking process, recycled paper, newsprint, and cardboard were employed as raw materials. The WWTP has a capacity of approximately 1.200 $m^3$ treats/day of the industrial effluent with a daily biological load of around 6.720 kg chemical oxygen demand (COD)/d using a BIOPAQ-UASB (Upflow Anaerobic Sludge Blanket, Veolia-Spain) anaerobic reactor combined with a classic aerobic treatment system. The sewage first passes through anaerobic ponds and then activated sludge ponds with anoxic and aeration zones.

The feed water samples were then stored in a refrigerator at 4 °C and warmed to room temperature prior to use in the ultrafiltration experiments. Average concentrations of constituents in the feed solution are summarized in Table 1.

**Table 1.** Average concentrations of major constituents in the feed water.

| Parameter | Units | Value | Equipment |
|---|---|---|---|
| Turbidity | (NTU) | 39.5 | Turbidimeter D-112—Dinko. |
| Chemical oxygen demand (COD) | (mg/L) | 252 ± 5.0 | Reactor 5B-2C, COD meter. |
| Total organic carbon (TOC) | (mg/L) | 80.00 ± 2.4 | TOC-VCSN Shimadzu Analyzer |
| Ultraviolet absorbance ($UV_{254}$) | ($cm^{-1}$) | 0.943 ± 0.012 | UV-VIS Scanning spectrophotometer (Unicam, UV2). |
| Suspended solids (TSS) | (g/L) | 0.1986 ± 0.05 | TSS Vaccum filtration assembly—Alamo/Dinko. |
| Sediment solids (SS) | (ml/L) | 3.5 ± 0.1 | Imhoff sedimentation cone—1000 mL—VITLAB. |
| Conductivity | (ms/cm) | 3.56 ± 1.0 | Conductivity Meter, EC-Metro GLP 31—Crison. |
| Total nitrogen | (mg/L) | 1.7 ± 0.2 | Thermoreactor AL125 and protometer—Lovibond. |
| Particle size | nm | 458–1281 | Zetasizer Nano ZS—Malvern Instruments. |

### 2.2. Membranes and UF Pilot Plant

UF experiments were carried out using PES membranes with different molecular weight cut-offs (MWCO) of 10 kDa, 50 kDa, and 100 kDa in a flat sheet with a surface area of 41.8 $cm^2$, provided by Synder Filtration™ (Vacaville, CA, USA). These membranes have nominal pore sizes of 3.16–3.5 nm, 6.09–8.17 nm, and 7.67–12.29 nm, respectively, calculated based on the literature [25,26].

As the foulant fraction between 0.45 µm and 100 kDa is the major fraction contributing to the membrane fouling [27], PMTE was a pre-filtrated at 0.45 µm to eliminate larger suspended solids and higher molecular weight colloids. Then, filtration experiments were carried out in a 400 mL stirring cell (Amicon 8400, Merck-Millipore, Merck KGaA, Darmstadt, Germany) connected to a pressurized tank (800 mL) and operated in dead-end filtration mode to filter large volumes of sample. The UF setup is shown in Figure 2.

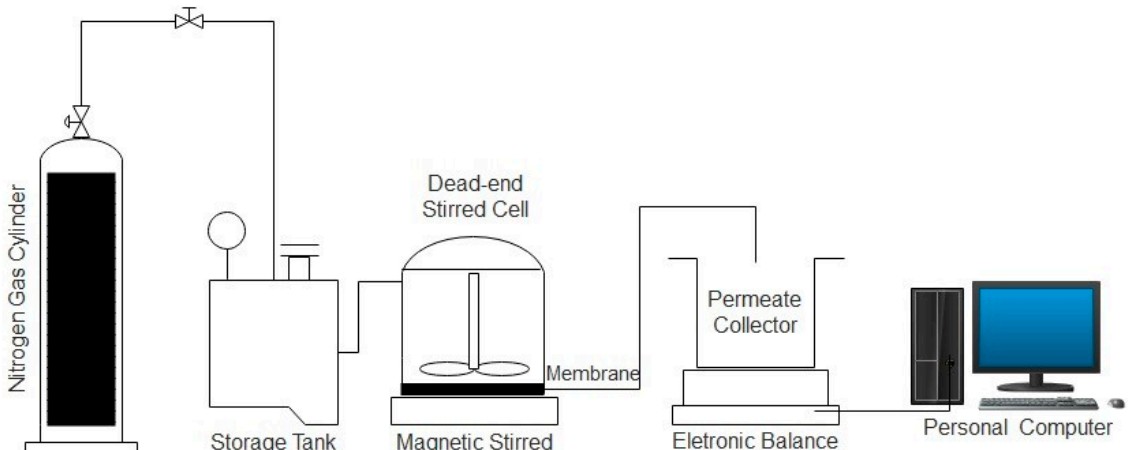

**Figure 2.** Process configuration for batch dead-end membrane filtration with stirred cell.

### 2.3. Membrane Filtration Tests

Before the UF experiments, the new membranes were soaked in Milli-Q water for 24 h and then carefully rinsed. For the membrane characterization measurements, distilled water was used as the feed solution for each membrane, and the water flux was given the term ($J_w$). Filtration were performed under different transmembrane pressures (TMPs) (0.5, 1.0, 1.5, and 2.0 bar) at room temperature.

After determining the clean water flux, the fouling experiments were performed on the different membranes MWCO (10 kDa, 50 kDa, and 100 kDa) to understand the role this parameter plays in fouling propensities. Every UF experiment included filtration of 250 mL of the pre-filtered PMTE as a feed sample. The filtration protocol also included membrane cleaning steps in accordance with the methodology described by C. Jacquin et al. [20] and set out below.

Dead-End Filtration Protocol

1. Pre-filtration: Filtration of 600 mL of the PMTE at 2.5 bar at 0.45 μm to eliminate the larger suspended solids and higher molecular weight colloids.
2. First filtration step: Filtration of 250 mL of pre-filtrated raw feed at 2.0 bar to understand the flux decline and fouling resistance behavior. A new membrane was cut and used for each filtration test.
3. Cleaning by relaxation: The filtration cell was refilled with 15 mL of buffer solution (NaHCO$_3$ 1 mmol/L) and stirred for 10 min at 100 rpm to remove foulants by simulating membrane relaxation.
4. Second filtration step: Filtration of 40 mL of buffer solution at 2.0 bar to calculate flux recovery and membrane resistance after membrane relaxation (reversible fouling).
5. Cleaning by backwashing: The membrane was turned upside down, and filtration with 30 mL of buffer solution at 2.0 bar was carried out to remove foulants by performing a membrane backwash.
6. Third filtration step: The membrane was put back in place, and the filtration of 30 mL buffer solution at 2.0 bar was performed to calculate the flux recovery.

### 2.4. Ultrafiltration Fouling Models

Permeate flux was gravimetrically measured at different time intervals, and the resistance-in-series model was used to study the role of different fouling resistances that cause flux decline on the membrane as described by Darcy's Equation [28]:

$$J_p = \frac{1}{A_m \, \rho} \frac{dm_p}{dt} = \frac{\Delta P}{\mu \left( R_m + R_f \right)}, \tag{1}$$

where $J_p$ is the permeate flux (L·m$^{-2}$·h$^{-1}$), $A_m$ is the effective membrane area (m$^{-2}$), $m_p$ is the total mass of permeate (kg), $\rho$ is the volumetric mass density (kg·m$^{-3}$), $t$ is the filtration time (s), $\Delta P$ is

transmembrane pressure drop (Pa), $\mu$ is the filtrate viscosity (Pa·s), $R_m$ is the intrinsic membrane resistance (m$^{-1}$) and $R_f$ is the fouling resistance (m$^{-1}$).

In dead-end filtration, and according to the cake filtration model, the relationship between $R_f$ and the specific cake resistance ($\alpha$) (m·kg$^{-1}$) [28,29], can be expressed by Equation (2):

$$R_f = \alpha M = \frac{\alpha \times c \times V_p}{A},$$ (2)

where

$M$ is the cake mass of DCS ($m_{DCS}$) per unit membrane area (kg·m$^{-2}$), and

$V_p$ is the permeate volume (m$^3$).

A mass balance equation is then used to obtain $c$ by the expression:

$$c = \frac{m_{DCS}}{V_p} = \frac{\left(V_0 c_0 - V_p c_p\right) - V_w c_w}{V_p},$$ (3)

where

$m_{DCS}$ is the cake mass of DCS (kg);

$c$ is the mass of solids deposited on the cake layer per unit volume of filtrate (kg/m$^3$);

$c_0$ and $c_p$ are the initial concentration in the feed and permeate concentration (kg/m$^3$), and

$V_0$ and $V_w$ are the initial volume to be filtered and fluid volume retained in the cake (m$^3$), respectively.

Therefore, the classic method for determining $\alpha$ is to measure $V_p$ as a function of time, $t$, during dead-end UF at constant pressure. Specific cake resistance can be calculated from the well-known expression derived using (Equation (1) and Equation (2)) and solving the follow integration [30]:

$$\int_0^t dt = \int_0^{V_p} \left( \frac{\mu \times R_m}{\Delta P \times A_m} + \frac{\mu \times \alpha \times c}{\Delta P \times A_m{}^2} V_p \right) dV_p.$$ (4)

This integrated result can be rearranged to give the linear equation, where $\alpha$ can be calculated from the slope by a simple plot of $t/V_f = f(V_f)$ at different applied pressures [31].

In addition, [32] proposed a mathematical model Equation 5 denominated pore-blocking fouling mechanism or Hermia's law to describe permeate flux decline phenomena, based on classical constant-pressure of dead-end filtration.

$$\frac{d^2 t}{dV_f{}^2} = K\left( \frac{dt}{dV_f} \right)^m.$$ (5)

The exponent $m$ characterizes the type of blockage for each fouling mechanism, while the value of $K$ represents a constant fouling parameter that varied for each filtration processes. Hence, pore blocking during filtration can be divided into four mechanisms: complete blocking ($m = 2$), intermediate blocking ($m = 1$), standard blocking ($m = 1.5$), and cake layer formation ($m = 0$).

The analytical solutions of Equation 5 for each $m$ value, as well as the linear forms of permeate flux models, are shown in Table 2 [32–34].

**Table 2.** Summary of the fouling mechanisms by blocking models during dead-end filtration.

| Fouling Mechanism | $m$ | Fouling Concept | Fouling Models | Linear Forms | |
|---|---|---|---|---|---|
| Complete blocking | 2 | Pore sealing | $J_p = J_0 exp(-K_{cb}t)$ | $ln\left(J_p\right) = ln(J_0) - K_{cb}t$ | (6) |
| Intermediate blocking | 1 | Pore sealing and membrane deposition | $J_p = \frac{J_0}{(1 + J_0 K_{ib}t)}$ | $\frac{1}{J_p} = \frac{1}{J_0} + K_{ib}t$ | (7) |
| Standard blocking | 1.5 | Pore walls restricted | $J_p = \frac{J_0}{\left(1 + J_0^{\frac{1}{2}} K_{sb}t\right)^2}$ | $\frac{1}{J_p^{\frac{1}{2}}} = \frac{1}{J_0^{\frac{1}{2}}} + K_{sb}t$ | (8) |
| Cake formation | 0 | Cake layers on surface | $J_p = J_0\left(1 + J_0^2 K_{cf}t\right)^{\frac{1}{2}}$ | $\frac{1}{J_p^2} = \frac{1}{J_0^2} + K_{cf}t$ | (9) |

Furthermore, as described in the filtration protocol, at the end of the UF and after the cleaning steps, the flux decline and flux recovery for each MWCO was estimated using Equations (10) and (11), sequentially:

$$Flux_{decline}(\%) = \left(1 - \frac{J_p}{J_0}\right) \times 100,$$ (10)

and

$$Flux_{recovery}(\%) = \frac{J_{p(AW)}}{J_0} \times 100,$$ (11)

where

$J_0$ is the initial permeate flux, and

$J_{p(AW)}$ is the permeate flux after cleaning.

### 2.5. Mass Balance Analysis

To determine the $m_{DCS}$, a mass balance was performed for the three different membranes (10 kDa, 50 kDa, and 100 kDa) fouled by PMTE. The mass balance for the solution was calculated using the following equation:

$$m_{solution} = COT \times V_{solution}$$ (12)

where,

$m_{solution}$ is the mass of carbon in the solution (mgC);

$COT$ is the total organic carbon concentration measured in the solution (mgC/L), and

$V_{solution}$ is the volume of the solution (L).

After the cleaning steps (relaxation + backwashing), TOC measurements were performed for each collected solution (feed, permeate, relaxation, and backwashing) and the residual carbon mass on the membrane, termed irreversible carbon, was estimated using Equation (13).

$$m_{DCS\ irreversible} = m_{feed} - m_{permeate} - m_{relaxation} - m_{backwashing}$$ (13)

where,

$m_{DCS\ irreversible}$ is the carbon mass remaining on the membrane or irreversible fouling (mgC);

$m_{feed}$ is the carbon mass in the feed solution (mgC);

$m_{permeate}$ is the carbon mass collected in the permeate (mgC);

$m_{relaxation}$ is the carbon mass collected in the relaxation solution (mgC), and

$m_{backwashing}$ is the carbon mass collected in the backwashing solution (mgC).

The total organic matter (TOC) was measured using a TOC-VCSN Shimadzu Analyzer (Shimadzu Europa GmbH, Duisburg, Germany). TOC samples were filtered through a 0.45 μm filter before analysis.

### 2.6. Specific Ultraviolet (UV) Absorbance (SUVA$_{254}$)

The specific UV absorbance (SUVA) is determined by the ratio of the UV absorbance measured at 254 nm to TOC concentration [35,36]. SUVA was calculated according to the following equation:

$$SUVA\left(\frac{L}{m \times mg}\right) = UV_{254}(m^{-1}) \times \frac{100}{TOC\left(\frac{mg}{L}\right)}$$ (14)

where high SUVA values (more than 4 L/m/mg) indicate high humic content with a hydrophobic character (aromatic), and low SUVA (less than 3 L/m/mg) corresponds to the presence of mainly hydrophilic material with reduced aromatic character [37,38]. Specific UV absorbance (SUVA$_{254}$) was measured with a 1 cm quartz cuvette using a UV-VIS spectrophotometer (UV-2401PC, Shimadzu, Kyoto, Japan). Samples were measured in triplicate, and the results were averaged.

### 2.7. Field Emission Scanning Electron Microscopy (FESEM) and Energy Dispersive Spectrophotometry (EDS)

Micrographic imaging and elemental analysis of the fresh and fouled membranes were undertaken using FESEM in conjunction with EDS. FESEM analysis gives a qualitative assessment of the fouling formed on the membrane surface. EDS determines the inorganic composition of the foulants. The dried membrane samples were attached to double-sided adhesive carbon tape on an aluminum holder, and subsequently coated with platinum and observed by FESEM using a Hitachi S-4500 apparatus (Hitachi High-Tech Fielding Corporation, Fukuoka, Japan), at an accelerating voltage of 0.5–30 kV, and a working distance of 10 mm. Inorganic foulants on the membrane were determined by EDS (Aztec EDS with X-Max detector, Oxford Instruments plc, Abingdon, UK). The qualitative and quantitative analyses of EDS spectra were based on internal standards using Aztec software.

### 2.8. Attenuated Total Reflection-Fourier Transform Infrared (ATR-FTIR) Spectroscopy Analysis

The chemical structure of the clean, fouled, and cleaned membrane samples (10 kDa, 50 kDa, and 100 kDa) were analyzed using an FTIR spectrometer in attenuated total reflectance (ATR) mode (Nicolet™ FT-IR spectrometers from Thermo Electron Corporation, with Universal ATR Sampling Attachment, Waltham, MA, USA). The resolution of the ATR-FTIR apparatus was attuned to 4.0 cm$^{-1}$, optical path difference velocity to 0.2 cm/s, and all the spectra were recorded within the range 4000–400 cm$^{-1}$. A ZnSe crystal (thallous bromide iodide) was used as an internal reflection element. The effective incident angle of the IR radiation was 45°. The background air spectrum was subtracted, and the spectra were offset corrected, normalized, and presented in absorbance. All the membranes were dried by slow evaporation in a desiccator overnight at room temperature prior to ART-FTIR characterization to minimize interference from water bands. The analysis was conducted at three random points on the membrane surface.

### 2.9. Foulant Extraction

The foulants were carefully extracted from the membranes using the following steps adapted from previous studies by [39,40]:

The fouled membrane samples of 41.8 cm$^2$ were dried in a desiccator at room temperature overnight and then cut into small pieces (approximately 1 cm$^2$).

Then, the membrane samples were placed into a 50 mL glass bottle and soaked in a 20 mL acetone–water solution (1:9 v/v) for 24 h at 20 °C and completely mixed using a magnetic blender to ensure all the membrane pieces were in contact with the solvent.

The extract solution, consisting of a mixing of DCS, was filtrated through a syringe filter (membrane) with a mean pore size of 0.22 μm.

The filtrate was placed in a round-bottom flask and evaporated to dryness at a temperature of approximately 50–55 °C under a vacuum pressure of around 146 mbar for 30 min, using a Buchi rotavapor R-114 extraction system (Büchi Labortechnik AG, Flawil, Switzerland) with a PC 600 dry pumping chemistry vacuum unit (Vacuubrand, Wertheim, Germany).

Finally, the foulant was re-dissolved in 3 mL of the acetone–water solution (1:9 v/v) and then subjected to further 3DEEM and other analyses.

### 2.10. DEEM Fluorescence Spectra Analysis

3DEEM analyses were performed using a Perkin-Elmer, LS-55 fluorescence spectrometer (PerkinElmer Inc., Waltham, MA, USA) at room temperature (22.0 ± 2 °C). In this study, 3DEEM spectra were collected with the scanning excitation wavelength ($\lambda_{ex}$) set at 200–500 nm and the emission wavelength ($\lambda_{em}$) from 280 nm to 600 nm. Scan speed was set at 1000 nm/min and the increment to 2 nm, while the slit width was set at 10 nm in excitation and emission [20,21]. To avoid Raman scatter by the particles [19], fluorescence measurements were done on pre-filtered (0.45 μm) at room temperature (22 ± 2 °C). To limit overlapping signals and avoid the inner filter effect, the samples

were diluted with pure water (Mili-Q, Millipore, Merck KGaA, Darmstadt, Germany) with a dilution ratio determined after measurements at successive dilution ratios [20,41,42]. All spectra were Raman normalized using a Mili-Q water blank (the Milli-Q water spectrum was subtracted from the 3DEEM spectrum for each sample) following the procedure described by Peiris et al. [22] and Goletz et al. [43].

Therefore, after Raman normalization, the fluorescence intensity in Raman units (R.U) was assessed [44]. As described in a previous study carried out by W. Chen et al. [21], the spectra were divided into five areas (Table 3).

**Table 3.** Characteristics of the associated fluorophores detected by a 3D fluorescence excitation–emission matrix (3DEEM), according to W. Chen et al. [13].

| Region | Associated Fluorophores | Excitation and Emission Wavelengths | Abbreviation |
|---|---|---|---|
| Region I | Aromatic protein I-like (tyrosine) | Ex = 200–250 nm<br>Em = 280–330 nm | $Prot_1$—like$_{3DEEM}$ |
| Region II | Aromatic protein II-like ($BOD_5$) | Ex = 200–250 nm<br>Em = 330–380 nm | $Prot_2$—like$_{3DEEM}$ |
| Region III | Fulvic acid-like | Ex = 200–250 nm<br>Em = 380–600 nm | FA—like$_{3DEEM}$ |
| Region IV | Soluble microbial product (tryptophane) | Ex = 250–350 nm<br>Em = 280–380 nm | SMP—like$_{3DEEM}$ |
| Region V | Humic acid-like molecules | Ex = 250–500 nm<br>Em = 380–500 nm | HA—like$_{3DEEM}$ |

DCS fractions, and consequently 3DEEM data, were analyzed quantitatively using the volume of fluorescence $\Phi(i)$ (R.U.nm$^2$) parameter from the fluorescence regional integration (FRI) method originally proposed by W. Chen et al. [21]. This methodology permits all the fluorescent compounds detected within each fluorophore region (Table 3) to be considered. Fluorescence volumes were calculated from the corrected matrix, using the Equation (1) [21]:

$$\Phi(i) = MF(i) \sum_{ex} \sum_{em} I_{final}(\lambda_{ex}, \lambda_{em}) \Delta\lambda_{ex} \Delta\lambda_{em} \tag{15}$$

where $MF(i)$ is the area multiplication factor, $I_{final}(\lambda_{ex}, \lambda_{em})$ is the final fluorescence intensity at each excitation—emission wavelength in (R.U.).

$\Phi(i)$ normalization was necessary to compare values from different regions of the 3DEEM response. To make it, $MF(i)$ was calculated using Equation (2) [45]:

$$MF(i) = \frac{Area_{tot(3DEEM)}}{Area_{region(i)}}, \tag{2}$$

where
$Area_{tot(3DEEM)}$ is the total spectrum area (nm$^2$), and
$Area_{region(i)}$ = is the specific region area ($i$) (nm$^2$).

## 3. Results and Discussion

### 3.1. Analysis of the Influence of Membrane MWCO on Permeate Flux and Fouling Mechanism

One of the main goals of the filtration experiments was to understand the effects of MWCO on membrane fouling behavior. As shown in Figure 3, during the UF in dead-end mode at constant pressure, the initial permeate flux increased with an increase in MWCO due to the direct relationship between permeate flux and nominal membrane pore sizes. Thus, when comparing the three membranes used in the filtration experiments, the flux was higher for membranes with a larger MWCO.

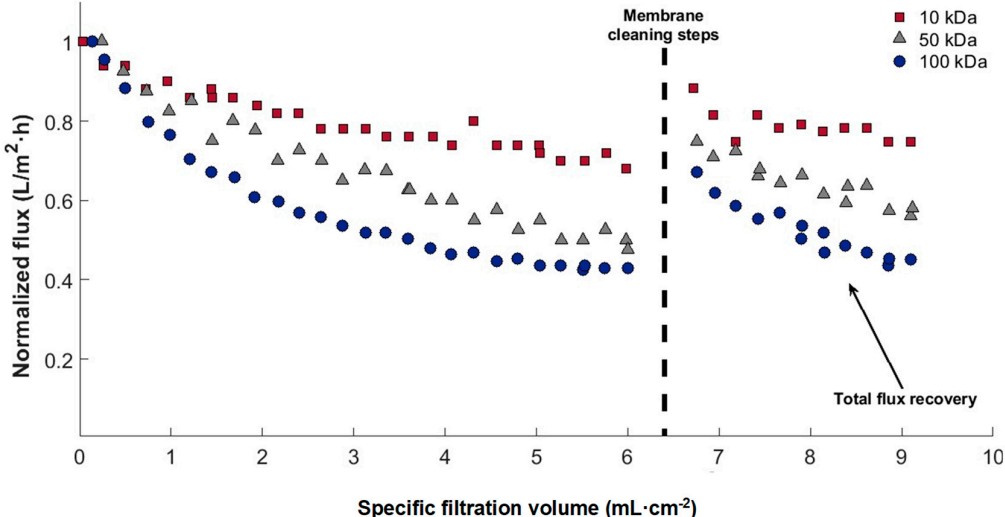

**Figure 3.** Flux vs. permeate volume, during filtration of PMTE using 10 kDa, 50 kDa, and 100 kDa ultrafiltration-polyethersulfone (UF-PES) membranes. The applied transmembrane pressure was 2.0 bar, at room temperature, and the total organic carbon (TOC) concentration of the feed solution was about $78.0 \pm 2.5$ mg·L$^{-1}$.

The initial permeate flux was found for the 100 kDa membrane with $J_0 = 102.77$ L/m$^2$/h followed by the 50 kDa membrane with $J_0 = 73.49$ L/m$^2$/h and then by 10 kDa membrane with $J_0 = 28.70$ L/m$^2$/h. However, it can be seen that even though the 100 kDa membrane presented the greatest initial flux, it declined very quickly and mainly during the first few minutes of the filtration, which can be attributed to the deposition of DCS and their adsorption into the membrane pores, followed by cake formation. Therefore, after filtration of 6.0 mL·m$^{-2}$ (i.e., $250 \pm 0.5$ mL of PMTE), it was noted that for the lowest MWCO (10 kDa), the flux decline was smaller and slower compared to the larger MWCO (50 kDa and then 100 kDa). The highest reduction in the permeate flux was 56.98% for the 100 kDa membrane, followed by 52.50% for the 50 kDa membrane and 32.50% for the 10 kDa membrane.

A very significant membrane flux recovery was observed immediately after cleaning for all MWCO (dead-end filtration protocol) (see Figure 3), indicating high effectiveness in removing the cake layer formed by the colloids associated with reversible fouling. The membrane performance and flux recoveries obtained from the different MWCO are reported in Table 4.

**Table 4.** Permeate flux, flux reduction, and total flux recovery after cleaning steps (relaxation and backwashing) from different molecular weight cut-off (MWCO) membranes.

| MWCO | Permeate Flux on Clean Membrane (L·m$^{-2}$·h$^{-1}$) | Flux Reduction (%) | Permeate Flux After Cleaning Step (L·m$^{-2}$·h$^{-1}$) | Total Flux Recovery (%) |
|---|---|---|---|---|
| 10 kDa | $28.70 \pm 1.23$ | 32.00 | $25.38 \pm 0.49$ | 88.40 |
| 50 kDa | $73.49 \pm 1.17$ | 52.50 | $54.89 \pm 1.04$ | 74.69 |
| 100 kDa | $102.78 \pm 2.65$ | 56.98 | $68.90 \pm 3.00$ | 67.03 |

*3.2. Resistance-in-Series and Pore Blocking Model Analysis*

In Figure 4, the results of filtration experiments for all UF membranes tested, are presented on the plot of $t/V_f$ vs. $V_f$, which should be linear when the fouling mechanism is the cake filtration by Equation (4). The resistance-in-series model Equation (1) and Equation (2) was used to analyze membrane resistances that lead to flux decline during the UF process. As shown in Figure 4, the fouling resistance ($R_f$) increased throughout the filtration period, due to pore blocking during the earlier stages, and cake growth on the membrane surface during the later stages of UF and $R_f$ was highest for the largest MWCO (i.e., 100 kDa), which indicates that more cohesive cake layers form on the membranes

and, consequently, fouling becomes more irreversible. These results are consistent with those presented by Lee S. [46].

The highest values for $R_f$ were obtained at the highest mass of DCS ($m_{DCS}$) deposited on the membrane area per unit of permeate volume due to a more compact cake structure on the membrane surface. Thus, the $R_f$ for the 10 kDa MWCO membrane, was around $8.90 \times 10^{13}$ $m^{-1}$, for the 50 kDa MWCO, it was $1.09 \times 10^{14}$ $m^{-1}$, and for the 100 kDa MWCO, it was $1.77 \times 10^{14}$ $m^{-1}$.

It is worth mentioning that the nonlinearity of the curves $t/V_f$ vs. $V_f$ during the early stages of experiments implies that pore-blocking preceded the cake resistance in the early stage of filtration. Therefore, pore-blocking mechanisms were identified with a method based on a simple parameter estimation in nonlinear regression models (Table 2). Table 5 presented the values of optimized parameters $K_{Cb}$, $K_{ib}$, $K_{sb}$, and $K_{cl}$ according to Equations (6)–(9), respectively, the comparison of experimental average permeate flux and the predicted average flux, and the corresponding correlation coefficients ($R^2$).

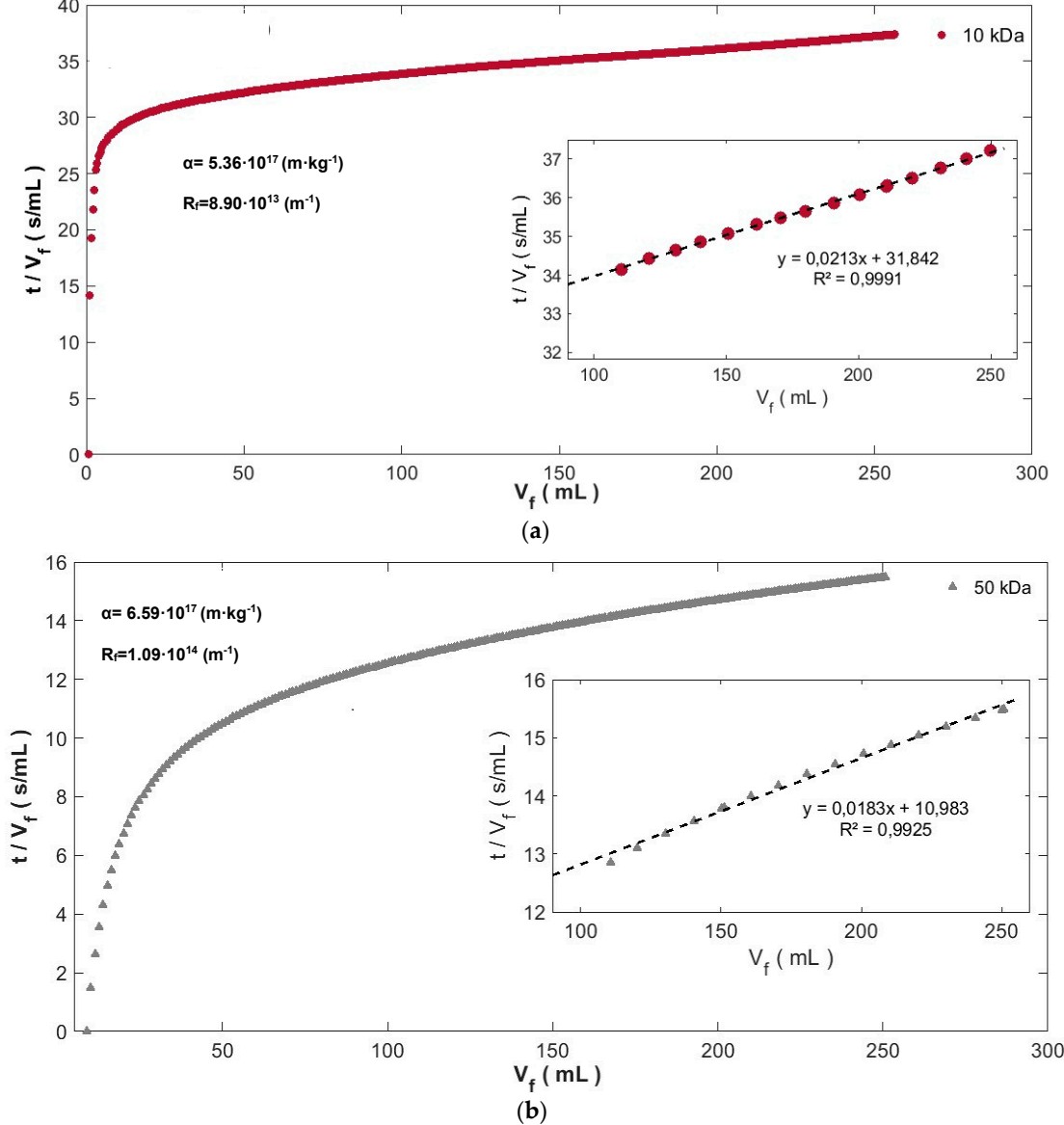

**Figure 4.** *Cont.*

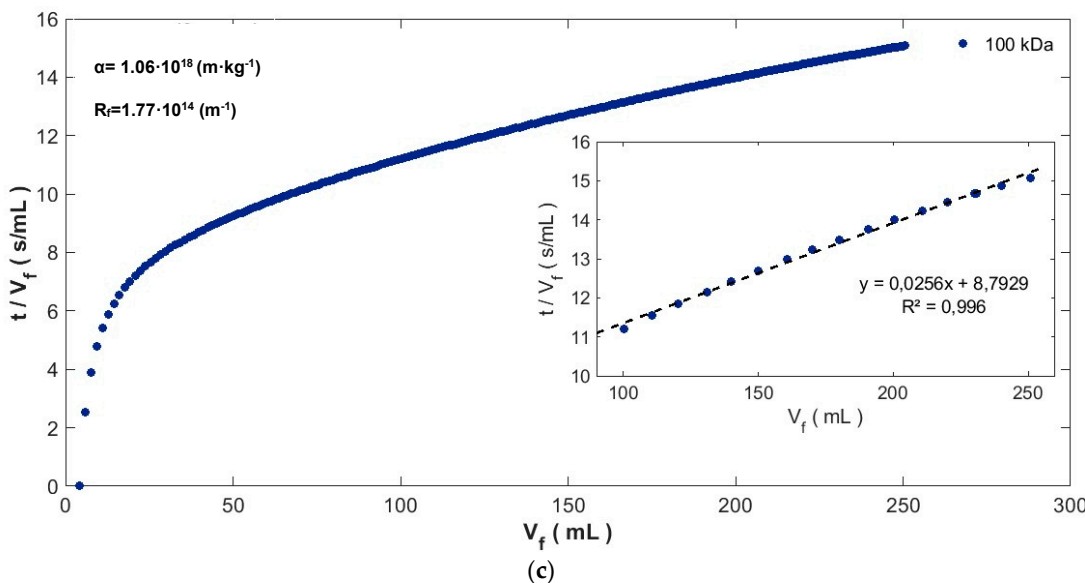

**Figure 4.** Specific resistance ($\alpha$) and fouling resistance ($R_f$) calculated using the resistance-in series-model for (**a**) 10 kDa molecular weight cut-off (MWCO), (**b**) 50 kDa MWCO, and (**c**) 100 kDa MWCO membranes.

**Table 5.** Values of pore-blocking parameters, comparison between the experimental and predicted average permeate flux, and the model fitting accuracy ($R^2$).

| Models | MWCO (kDa) | Constant Fouling ($K$) | $\bar{J}_{p(exp)}$ ($L \cdot m^{-2} \cdot h^{-1}$) | $\bar{J}_{p(pred)}$ ($L \cdot m^{-2} \cdot h^{-1}$) | $R^2$ |
|---|---|---|---|---|---|
| Cake filtration m = 0 | 10 | $2.06 \times 10^6$ | 23.071 | 23.277 | 0.878 |
| | 50 | $1.20 \times 10^6$ | 54.314 | 54.582 | 0.949 |
| | 100 | $1.99 \times 10^6$ | 59.961 | 59.939 | 0.969 |
| Intermediate blocking m = 1 | 10 | 6.94 | 23.071 | 23.376 | 0.824 |
| | 50 | 9.76 | 54.314 | 54.928 | 0.890 |
| | 100 | 18.61 | 59.961 | 60.628 | 0.875 |
| Standard blocking m = 1.5 | 10 | $9.0 \times 10^{-3}$ | 23.071 | 23.438 | 0.789 |
| | 50 | $19.6 \times 10^{-3}$ | 54.314 | 55.177 | 0.846 |
| | 100 | $39.7 \times 10^{-3}$ | 59.961 | 61.375 | 0.788 |
| Complete blocking m = 2 | 10 | $4.66 \times 10^{-5}$ | 23.071 | 23.509 | 0.749 |
| | 50 | $1.56 \times 10^{-4}$ | 54.314 | 55.494 | 0.790 |
| | 100 | $3.36 \times 10^{-4}$ | 59.961 | 62.569 | 0.662 |

$K$ unit depending on the parameter *m*. $K_{cb}$ ($s^{-1}$); $K_{ib}$ ($m^{-1}$); $K_{sb}$ ($m^{-1/2} \cdot s^{-1/2}$), and $K_{cf}$ ($s/m^2$).

As can be seen from Table 5, for all MWCO, cake formation has the closest values of experimental permeate flux and has higher $R^2$ values, between 0.878 (10 kDa) and 0.969 (100 kDa), confirming that the cake layer formation became the most dominant mechanism for permeate flux decline during UF of the PMTE. However, intermediate pore blocking played an important role in the fouling of membranes in the current study (see $R^2$ in Table 5). It indicates that the majority of foulants (macromolecules and colloids, such as proteins and polysaccharides) in the feed solution [47] have a bigger size than membrane pores. Those particles could not enter the membrane pores were retained on the membrane surface and, consequently, allow the formation and compression of the cake layer.

### 3.3. Mass Balance Analysis

The mass balance and TOC measurements were used to evaluate the irreversible carbon (fouling) deposited and/or adsorbed onto the membrane (see Figure 5) after the cleaning steps described in dead-end filtration protocol.

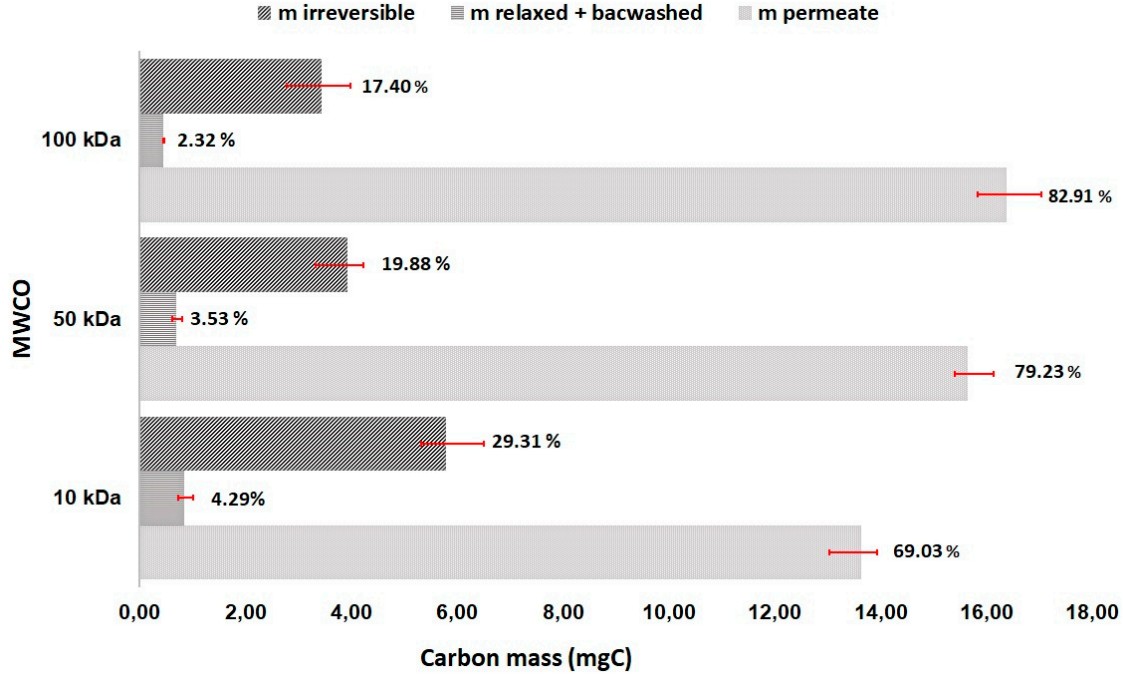

**Figure 5.** Carbon mass balances during UF for different MWCO, $m_{feed}$ = 19.76 ± 0.5 mg C.

It can be observed that due to DCS complexity and their large size distribution, an interaction between the particles, colloids, and dissolved matter presented in the feed solution with MWCO, thus membrane pore size played a key role in the organic carbon retained on the membrane. Consequently, the irreversible fouling in the mass balance increased with decreasing MWCO as present in Figure 5, showing disagreement with the results obtained in the membrane cleaning efficiency and flux recovery results presented in Table 4. According to the calculation by Equation (13), approximately 5.79 mg C, 3.93 mg C, and 3.44 mg C remained on the membranes (10 kDa, 50 kDa, and 100 kDa, respectively) as carbon mass irreversible ($m_{DCS\ irreversible}$).

### 3.4. Aromatic Carbon (SUVA) Removal by UF Membrane

In this study, the specific UV absorbance (SUVA) and TOC were used to assess the aromatic carbon and humic substance (HS) content of the DCS in the feed solution (PMTE) and permeates. Table 6 shows the SUVA values for the PMTE and the permeates at different MWCO. It can be seen that the SUVA values barely decrease in the permeates as compared with the feed solution. The mean SUVA value of the PMTE was 1.11 ± 0.03 L/mg/m, and this implies a low hydrophobicity and low molecular weight of the DCS in the effluent.

**Table 6.** Aromatic carbon (SUVA) in the PMTE and permeates, at 2.0 bar and different MWCO.

| | **Raw PMTE** | **Permeates** | | |
| --- | --- | --- | --- | --- |
| | | **10 kDa** | **50 kDa** | **100 kDa** |
| $UV_{254}$ (cm$^{-1}$) | 0.943 ± 0.012 | 0.518 ± 0.021 | 0.582 ± 0.010 | 0.648 ± 0.014 |
| TOC (mg·L$^{-1}$) | 80.00 ± 2.46 | 50.60 ± 1.72 | 61.21 ± 1.45 | 63.88 ± 1.35 |
| Reduction of COT | | 33.92% | 23.48% | 20.14% |
| SUVA (L·mg$^{-1}$·m$^{-1}$) | 1.114 ± 0.030 | 1.010 ± 0.045 | 0.956 ± 0.024 | 0.993 ± 0.026 |

As shown in Table 6, the mean SUVA values (feed and permeates) were relatively low (less than 3 L/mg/m), which suggest that most of the organic carbon material content in the PMTE consists mainly of hydrophilic components with low aromaticity and with low rejection by UF membranes. In addition, the TOC removal efficiencies of the 10 kDa, 50 kDa, and 100 kDa MWCO were relatively low (23.93%, 18.33%, and 12.24%, respectively).

### 3.5. DEEM Fluorescence Analysis

In this study, a 3D fluorescence excitation–emission matrix was used to identify and characterize the organic matter in the PMTE and permeates and on the fouled membranes. The 3DEEM fluorescence spectra for the PMTE and permeates for 10-kDa, 50-kDa, and 100-kDa PES membranes are shown in Figure 6. In addition, as described in Section 2.10, the spectra were analyzed into the five areas defined by W. Chen et al. [21].

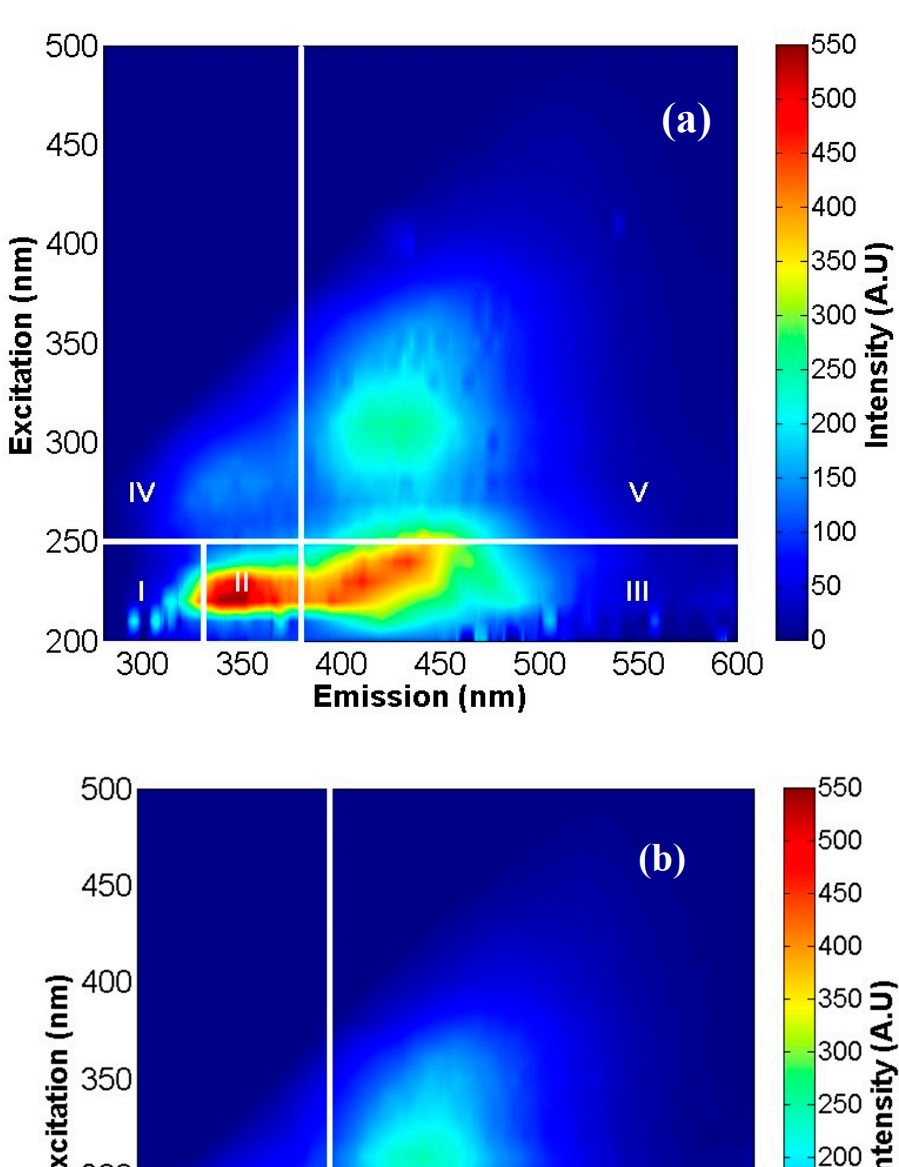

**Figure 6.** *Cont*.

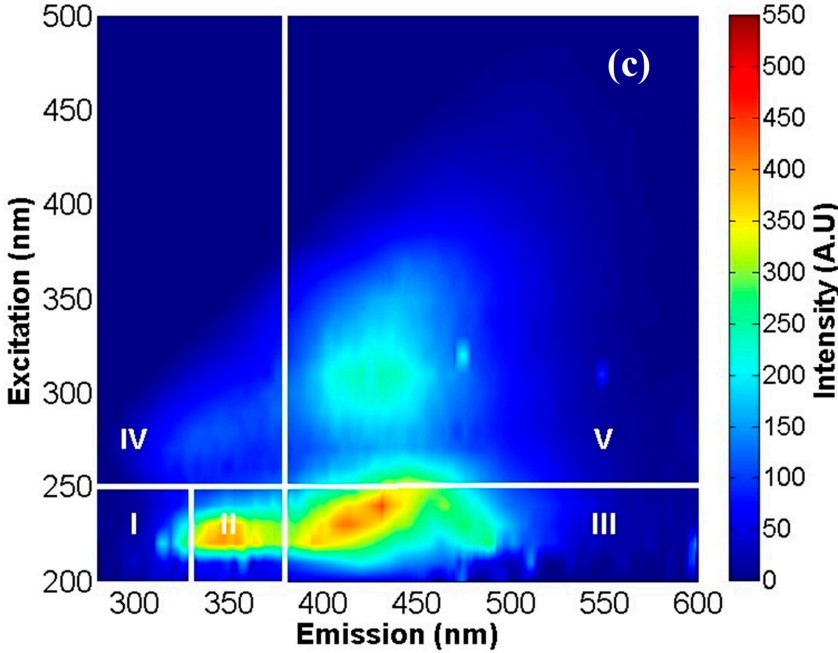

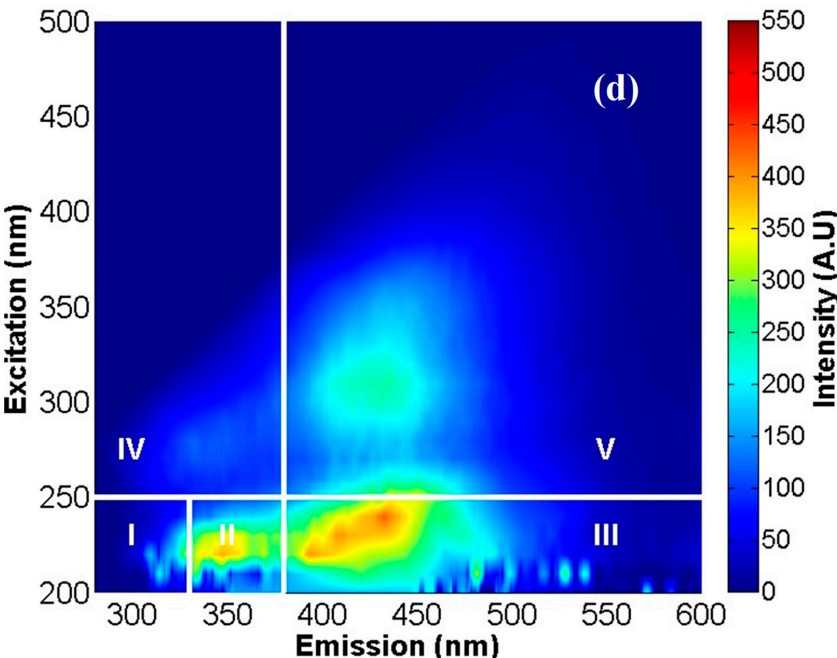

**Figure 6.** 3D fluorescence excitation–emission matrix (3DEEM) fluorescence spectra for (**a**) feed solution (prefiltered with 0.45 μm filter); (**b**) permeate 10 kDa; (**c**) permeate 50 kDa, and (**d**) permeate 100 kDa. Region I and II = aromatic protein-like substances I and II, respectively; Region III = fulvic acid-like substances; Region IV = soluble microbial by-products; Region V = humic acid-like substances.

In Figure 6, it can easily be seen from the qualitative analyses that aromatic protein-like substance II and fulvic acid-like substances are predominant in the feed solution. In addition, after dead-end UF at 2.0 bar, a decrease in the percentage fluorescence intensity in all membranes was observed, mainly in regions II, III, and IV. This suggests that a significant proportion of the fluorescent protein-like substances (high MW), including $BOD_5$, tryptophan, and extracellular polymeric substances (EPS), were removed in the permeates. Moreover, it can be seen that the decrease in the fluorophore compounds'

intensity was higher in the lower MWCO (10 kDa) due to the difference between the substances' molecular weight (MW) and the membrane pore size.

In addition, the 3DEEM data were analyzed quantitatively using the volume of fluorescence $\Phi$ (*i*) W. Chen et al. [21] and C. Jacquin et al. [20]. The volume of fluorescence and the reduction in fluorescent organic matter compounds are shown in Table 7.

**Table 7.** Volume of fluorescence $\Phi$ (*i*) and the reduction in the concentration of fluorescent compounds after ultrafiltration (UF).

| Region | Volume of Fluorescence (R.U.nm$^2$) | | | | Reduction of Fluorescent (%) | | |
|---|---|---|---|---|---|---|---|
| | Feed Solution | Permeate after UF | | | $\frac{\Phi (i)_{Feed} - \Phi (i)_{Permeate}}{\Phi (i)_{Feed}}$ | | |
| | | 10 kDa | 50 kDa | 100 kDa | 10 kDa | 50 kDa | 100 kDa |
| I | 2,950,264.07 | 1,853,482.64 | 1,982,014.02 | 2,196,593.78 | 59.17 | 48.85 | 34,.1 |
| II | 10,685,689.19 | 7,047,811.89 | 7,772,705.75 | 8,397,025.56 | 51.62 | 37.48 | 27.26 |
| III | 13,142,967.02 | 10,914,393.23 | 11,086,417.93 | 12,683,820.51 | 20.42 | 18.55 | 3.62 |
| IV | 5,173,347.62 | 4,167,024.60 | 4,317,331.47 | 4,465,922.64 | 24.15 | 19.83 | 15.84 |
| V | 4,170,485.25 | 3,918,418.88 | 3,877,028.25 | 4,013,670.38 | 6.43 | 7.57 | 3.91 |
| Total | 36,122,753.15 | 27,901,131.23 | 29,035,497.41 | 31,957,032.86 | 29.47 | 24.41 | 13.04 |

In addition to the results shown in Table 7, the membrane filtration process reduced the concentration of fluorescent compounds, mainly in regions I, II, and IV and it can be observed that the reduction in the concentration of fluorescent compounds increased when the MWCO decreased, which may confirm that the colloidal matter (i.e., particle size > 220 nm) plays a major role in membrane fouling. Moreover, depending on their size, this suggests that the main types of fouling are pore-blocking, followed by cake formation and growth during UF. In addition, in the graph of t/V versus V$_f$ (Figure 7), the linear trend confirms this assumption and makes it possible to calculate the colloid specific resistance.

Nevertheless, the quantitative volume of fluorescence in regions III and V (fulvic acids and humic acids) was not significantly decreased, as can be seen in Table 7, confirming their affiliation to hydrophilic and low MW compounds. These results were confirmed by the minimal SUVA removal in the permeates shown in Table 6, indicating that humic concentrations remained almost the same after the UF. It can also be seen that Fulvic-like and humic-like proteins (dissolved substances) predominated in the permeate with low-MW concentration.

To understand the role of organic matter (DCS) on membrane fouling, 3DEEM spectra of the foulants extracted from membranes fouled by PMTE are presented in Figure 7. This analysis was carried out in accordance with the one proposed by [45]. The organic matter was thus, combined into three regions of fluorophore groups:

- Associated with colloidal proteins (regions I + II) denominated by protein-like substances I+II;
- Associated with dissolved organic matter (region III + IV) termed fluvic acid-like and humic acid-like substances (FA+HA-like), and
- Associated with macromolecular proteins present in the dissolved phase (region IV) denominated by SMP-like substances.

It can be observed that the organic foulants with fluorescence characteristics extracted from the fouled membranes were significantly different from the PMTE and permeates (Figure 7). In addition, the fluorescence intensity of the 3DEEM spectra demonstrates that the colloidal proteins (protein-like substances I+II) and macromolecular proteins (SMP-like substances) were major fluorescent components on the fouled membranes, which agrees with previous studies carried out by [18,48,49], that reported one of the main compounds in membrane foulants as being proteins (tyrosine, BOD$_5$, and tryptophan).

The quantitative comparison of the fluorescent organic foulants on the fouled membranes was carried out by using the volume of fluorescence parameter from the fluorescence regional integration method. The distribution of $\Phi$ for the fouled membranes is presented in Figure 8.

In Figure 8, it can be observed that the $\Phi$ (*i*) distribution of the three different PES membrane was clearly different, which might be due to the different composition and retention in terms of fluorophore compounds on the fouled membranes. Hence, protein-like substances I+II comprised the fraction most retained by the membranes with approximately 62.71% retained by the 10 kDa membrane, around 64.75% for the 50 kDa membrane, and 63.93% for the 10 kDa membrane, followed by SMP-like molecules with around 20.55% retained by the 10 kDa membrane, 19.25 % by the 50 kDa membrane, and 19.63% by the100 kDa membrane. Moreover, it can be noted that the total $\Phi$ (*i*) increased as the MWCO decreased, 100 kDa to 50 kDa followed by 10 kDa. This foulant behavior can be explained by the hydrophobic characteristic and relatively higher MW or molecular mass in comparison with the membrane cut-off. However, FA+HA-like substances (strong hydrophobicity) were also found as a component of the membrane foulants, even though the dissolved organic matter was found in a lesser amount than colloidal proteins. In addition, they might be associated with irreversible fouling, due to their hydrophobic adhesion effect over and inside the membranes and as they affect both the hydraulic permeability and the rejection membrane properties [20,50,51].

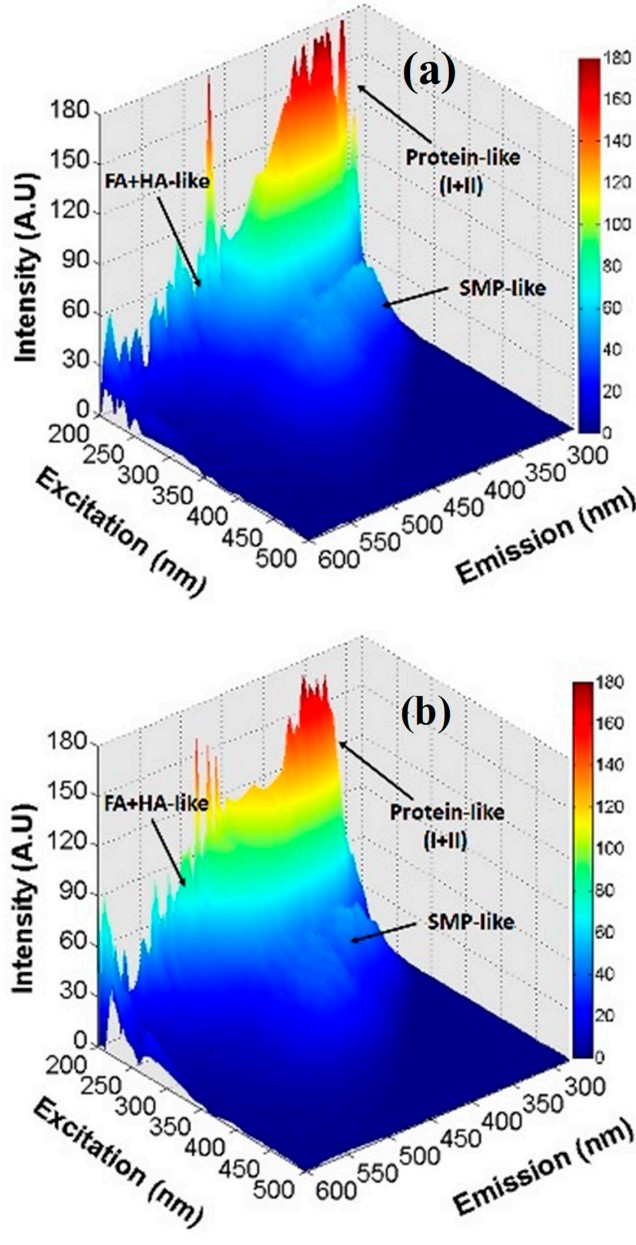

**Figure 7.** *Cont.*

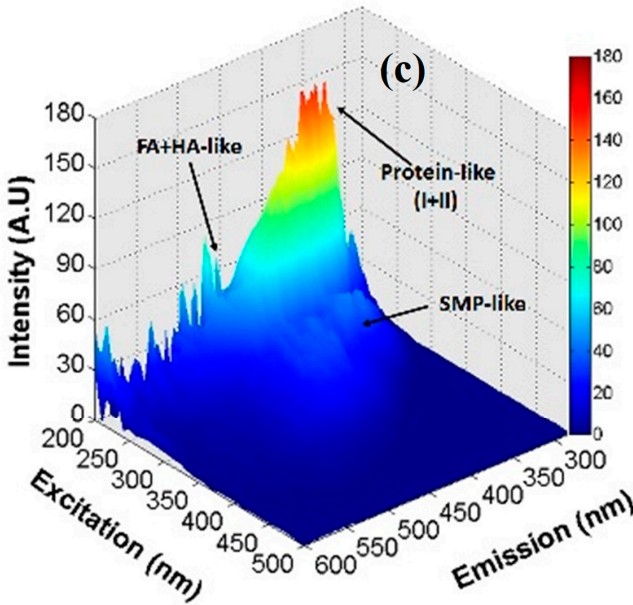

**Figure 7.** 3D view of fluorescence spectra of foulants extracted from fouled membranes at the end of filtration (dead-end filtration protocol): (**a**) 10 kDa; (**b**) 50 kDa, and (**c**) 100 kDa.

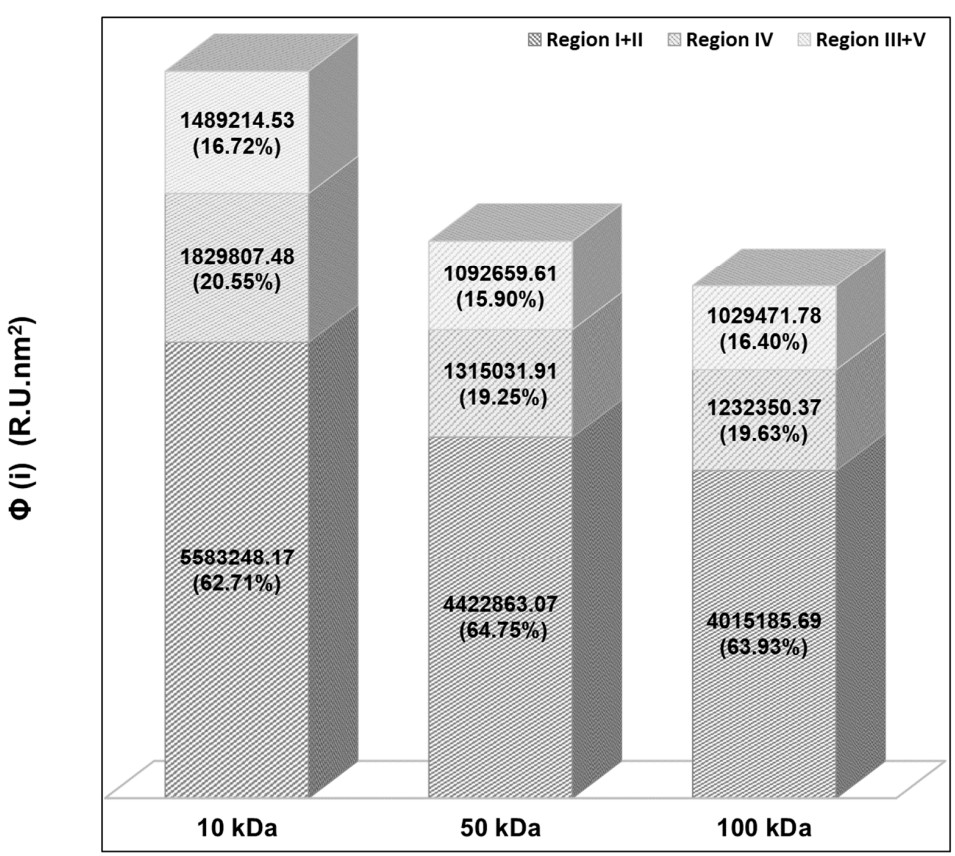

**Figure 8.** Volume of fluorescence distribution of foulants extracted from the fouled PES UF membranes.

## 3.6. ATR-FTIR Analysis

Fourier transform infrared spectroscopy was used to characterize and identify the major foulant groups, including the DCS, retained on the fouled membranes. ATR-FTIR is a rapid and reliable method used to detect the presence of different bands in the fouling layer (chemical functional groups), such as proteins, fatty and resin acids, colloids, and polysaccharides [52]. However, the chemical

complexity of paper mill effluents makes it difficult to be precise in characterizing membrane fouling using the FTIR test. Notwithstanding this, an important characteristic of all these foulant compounds was that they contain a C=O group in their structures, such as in the carboxylic acids [–C(=O)–OH] or carboxylate [–C(=O)–O$^-$], and this bond absorbs in a strong band in the range around 1690–1750 cm$^{-1}$ and 1550 cm$^{-1}$, respectively. In addition, all the carbohydrates absorbed at about 3400 cm$^{-1}$ (–C–OH) and at about 1060 cm$^{-1}$ (–C–OH or C–O–C) (Carlsson et al., 1998). Thus, this band was significant in IR (Infrared) fouling analysis of fouled membranes.

Several FTIR spectra from UF membranes (10 kDa, 50 kDa, and 100 kDa) were obtained in this study, the infrared spectra of the fresh PES UF membrane, fouled membranes by pre-filtered PMTE, and after cleaning procedures (dead-end filtration protocol: relaxation and backwashing steps) with NaHCO$_3$ (1 mmol/L) are provided in Figure 9.

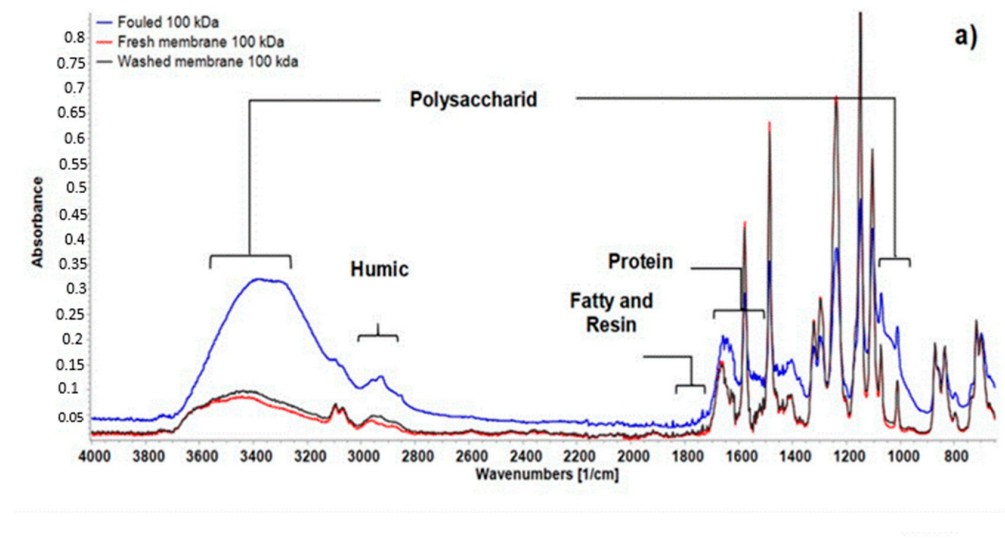

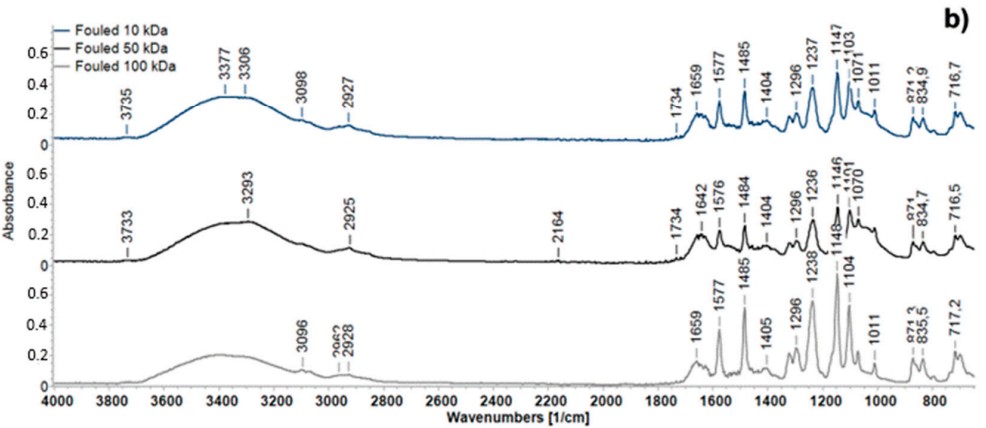

**Figure 9.** Attenuated total reflection-Fourier transform infrared (ATR-FTIR) spectra comparison of fresh PES membrane, fouled membrane, and after cleaning membranes procedures (**a**) (the IR spectrum comparison between MWCO membranes were almost the same. Therefore, only the spectra for 100 kDa membrane is shown) and DCS-fouled PES membranes (10 kDa, 50kDa and 100 kDa) by pre-filtered PMTE (**b**).

The broad bands between 3250 and 3400 cm$^{-1}$, were attributed to the overlapping of bands from the stretching vibrations of the N–H stretching in amides and the O–H stretching in the hydroxyl groups in the polysaccharides (DCS) within the membrane [53,54].

The bands between 2900 cm$^{-1}$ and 2850 cm$^{-1}$, which correspond to aliphatic-CH$_2$ asymmetrical stretching and symmetrical groups [55], and the bands near 1080 cm$^{-1}$ and 1070 cm$^{-1}$, which correspond to CH aromatics [56], both relate to the presence of humic substances.

The peaks absorbed at wavelengths around 1730 cm$^{-1}$ were indications of carboxylic groups attributed to fatty acids (carboxylic acid) and resin acids (carboxylate ion), both recognized contaminants in recycled paper mill effluent treated by the aerobic and anaerobic reactor, and they suggest a strong source of the membrane fouling [57].

The peaks located at 1650.55 cm$^{-1}$ and 1544.65 cm$^{-1}$ were due to C=O stretching in amide I and amide II attributed to the presence of proteins [49,58], suggesting they are a component of the EPS attached to the membrane surface.

The peaks around 1072 cm$^{-1}$ might suggest that polysaccharide-like substances (cellulosic species) were significant foulants on the membrane [59–61].

The identified functional groups and the typical organic compounds based on the IR spectra associated with the foulants in the membranes fouled by DCS coming from the paper mill treated effluent are shown in Table 8.

**Table 8.** Peaks and assignments of infrared spectra for clean and fouled membranes.

| Absorption Peak (cm$^{-1}$) | Associated Group | Compound |
|---|---|---|
| 1050 | C-O stretching and/or Stretching–S=O | Polysaccharide-like or Sugar ester sulfates |
| 1100–1080 | CH aromatic | Humic substances |
| 1570–1545 | Amide II (C—N and N-H bonds) | Proteins |
| 1670–1630 | Amide I (C=O) | Proteins |
| 1730 | [–C(=O)–OH] | Fatty acids |
| 2900–2850 | aliphatic–CH$_2$ stretching | Humic substances |
| 3400–3330 | Bonded N–H/C–H/O–H stretching vibration mode | Proteins, Polysaccharides and Humic substances |

Interpretation of IR spectra was based on Carlsson et al. [62], Ramamurthy et al. [57], and Puro et al. [40].

Therefore, according to the results, it can be suggested that the membrane foulants mainly consisted of fatty and resin acids, proteins, humic substances, and polysaccharides (cellulosic species), which is also consistent with the findings of the 3DEEM analysis. Then, the comparison of the signals between fouled and cleaned membrane shows that the irreversible foulant was aliphatic-CH$_2$ and bonded N–H/C–H/O–H stretching (see Figure 9a).

*3.7. FESEM and EDS Analysis*

Images of the fresh and fouled membrane structures were taken using field emission scanning electron microscopy (FESEM). Furthermore, energy dispersive spectroscopy (EDS) analyses were carried out to investigate the inorganic composition of the foulants deposited on the membranes.

Figure 10 shows the surface of the fresh, fouled, and cleaned PES membranes (10 kDa MWCO, 50 kDa MWCO, and 100 kDa MWCO). The pore can be easily identified before the UF. However, as expected, in the FESEM-images of the fouled membranes, a strong fouling layer could be seen on the membrane, due to the deposition of DCS, causing adsorption and pore blockage by low-molecular-weight contaminants and the formation of a cake layer by macromolecular contaminants accumulated on the membrane surface.

EDS analysis was, therefore, performed to investigate the inorganic foulants' composition and to study the influence of metal ions on membrane fouling. The elemental composition (EDS results) of both membranes (fresh and fouled) is shown in Table 9.

The EDS analyses confirmed that carbon, oxygen, and sulfur were the main elements detected in the fresh PES membranes, which agrees with the elemental chemical composition of polyethersulfone. In addition, it was easy to see the presence of most metal elements (sodium, calcium, magnesium, and silicate) on the fouled membranes, especially calcium, due to inorganic foulants [63].

In addition, the C, O, and S concentrations (weight %) were found to be different between the fresh and fouled membranes. The ratios of the different metal ions (Na$^+$, Mg$^{2+}$, SiO$_4^{4-}$, Cl$^-$, Ca$^{2+}$, K$^+$, Al$^{3+}$) on the fouled membrane surfaces were found to be different for the different MWCOs (10 kDa, 50 kDa, and 100 kDa).

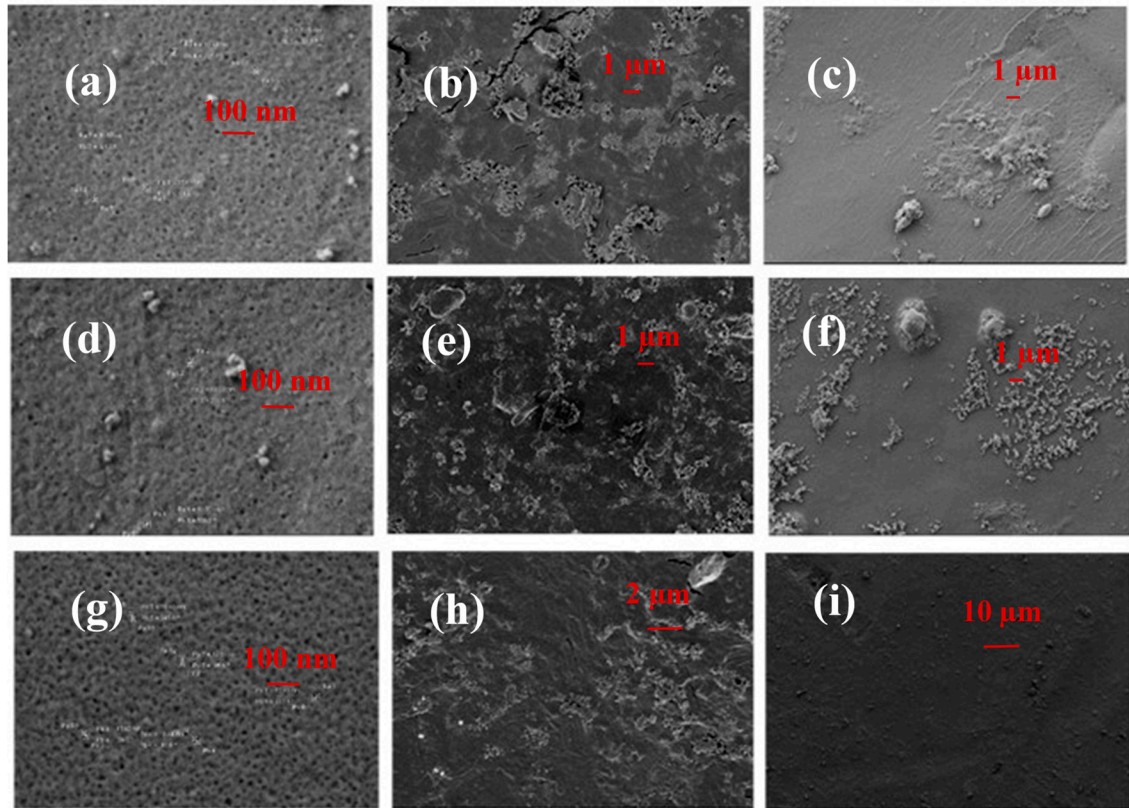

**Figure 10.** Field emission scanning electron microscopy (FESEM) images of the membrane surfaces (**a**) fresh 10 kDa MWCO, (**b**) fouled 10 kDa MWCO, (**c**) cleaned 10 kDa MWCO; (**d**) fresh 50 kDa MWCO, (**e**) fouled 50 kDa MWCO, (**f**) cleaned 50 kDa MWCO; (**g**) fresh 100 kDa MWCO, (**h**) fouled 100 kDa MWCO, (**i**) cleaned 100 kDa MWCO.

**Table 9.** Inorganic composition of fresh and fouled membranes.

| Element | 10 kDa | | | 50 kDa | | | 100 kDa | | |
|---|---|---|---|---|---|---|---|---|---|
| | Fresh | Fouled | Cleaned | Fresh | Fouled | Cleaned | Fresh | Fouled | Cleaned |
| | Weight (%) | Weight (%) | Weight (%) | Weight (%) | Weight (%) | Weight (%) | Weight (%) | Weight (%) | Weight (%) |
| C | 68.89 | 69.62 | 69.09 | 41.10 | 49.13 | 63.88 | 71.67 | 69.60 | 67.76 |
| O | 18.60 | 18.53 | 18.72 | 33.14 | 29.36 | 19.47 | 16.39 | 18.33 | 18.47 |
| S | 12.50 | 11.84 | 12.18 | 4.28 | 3.70 | 9.70 | 10.53 | 9.75 | 3.70 |
| Na | - | - | - | 2.77 | 4.87 | 2.15 | 0.34 | 0.77 | 1.92 |
| Mg | - | - | - | 1.08 | 0.63 | 0.16 | - | - | - |
| Si | - | - | - | 1.03 | 3.63 | 0.08 | - | - | - |
| Cl | - | - | - | 1.22 | 2.17 | 1.09 | 0.58 | 0.70 | 1.99 |
| K | - | - | - | 0.43 | 0.56 | 0.17 | 0.37 | 0.28 | 0.75 |
| Ca | - | - | - | 14.50 | 5.95 | 3.30 | 0.13 | 0.52 | 0.55 |
| Al | - | - | - | 0.44 | 0.41 | - | - | 0.04 | - |
| Total: | 100 | 100 | 100 | 100 | 100 | 100 | 100 | 100 | 100 |

It is worth mentioning that, as described by Chen et al. (2015) [7], the interaction between the multivalent metal ions (electrolytes), such as $Ca^{2+}$ and $Mg^{2+}$, can alter the thermodynamic and the kinetic stability of DCS via a Ca-and Mg-DCS complex and aggregate formation [64,65]. This consequently contributes to pore blocking and cake formation on the membrane surface during UF caused by the complexing of rejected colloids and metal ions in the PMTE, which was confirmed through EDS analysis of $Cl^-$, $K^+$, and $Ca^{2+}$ on washed membrane while Mg and Si were not detected.

Moreover, Carlsson et al. [54] described that, under alkaline conditions, fatty and resin acids (foulant compounds identified by ART-FTIR) are probably present in the form of calcium salts and a small fraction of them could coagulate and deposit on the membranes and consequently increase pore blocking.

## 4. Conclusions

Three different MWCO (PES) membranes were studied to estimate the degree of fouling caused by dissolved and colloidal matter (DCS), which come from paper mill treated effluent.

- Thus, it can clearly be seen that for the same volume of permeate, the fouling resistance during the filtration of PMTE increased when the MWCO increased from 10 kDa to 50 kDa and 100 kDa. This suggests that larger pore size induces higher flow resistance probably due to higher standard and complete blocking, which was confirmed by the trend of fouling constant $K_{SB}$ and $K_{CB}$, resulting in a greater degree of fouling. For all membranes, cake formation follows by an intermediate pore-blocking mechanism were the largest contributor to the observed permeate flux decline during the UF. In addition, it was observed that the relation between particle-size distribution in the feed solution and membrane pore size played a key role in the organic carbon retained on the membrane and consequently irreversible fouling.
- The 3DEEM analysis found that the dominant fluorescent substances on the fouled membranes were mainly associated with colloidal proteins and macromolecular proteins present in the dissolved phase as soluble microbial by-product-like materials, which might be explained by the protein-like substances I+II and the SMP-like substances in the DCS having a higher molecular weight than the MWCO and strong hydrophobic adhesion over the membrane pores, meaning they were consequently retained by the fine membrane pores and played a major role in the fouling on polyethersulfone UF membranes, whereas fluvic acid-like and humic acid-like substances were of lesser relevance.
- The ATR-FTIR and 3DEEM results agree with previous studies carried out by C. Jacquin et al. [20,45], Puro et al. [40], and Ramamurthy et al. [57]. So, it can be concluded that hydrophobic substances with large molecular weight, such as protein-like substances and polysaccharides, are mostly responsible for UF membrane fouling, whereas humic substances, which account for the majority of the dissolved organic matter in DCS, played a minor role. Therefore, the deposition and adsorption of proteins and polysaccharides during UF should be controlled by optimizing operational conditions, such as transmembrane pressure (TMP), cross-flow velocity (CFV), temperature and molecular weight cut-off (MWCO).
- FESEM and EDS analyses indicated that the foulants accumulated and adsorbed onto the membrane surfaces comprised not only organic matter but also inorganic elements including Na, Mg, Si, Cl, Ca, K, and Al. These results showed that the presence of multivalent metal ions, especially $Ca^{2+}$, on the fouled membrane can accelerate membrane fouling and can also contribute to irreversible fouling, whereas Mg and Si induce reversible fouling.

**Author Contributions:** Conceptualization, J.L.-G., M.-F.L.-P., M.R.S.S. and M.H.; methodology, J.L.-G., M.-F.L.-P., M.R.S.S. and M.H.; software, M.R.S.S.; validation, J.L.-G., M.-F.L.-P., M.R.S.S. and M.H.; formal analysis, J.L.-G., M.-F.L.-P., M.R.S.S. and M.H.; investigation, M.R.S.S. and M.-F.L.-P.; resources, M.R.S.S.; data curation, J.L.-G., M.-F.L.-P. and M.H.; writing—original draft preparation, M.R.S.S.; writing—review and editing, M.-F.L.-P. and J.L.-G.; visualization, M.R.S.S. and M.-F.L.-P.; supervision, J.L.-G. All authors have read and agreed to the published version of the manuscript.

**Funding:** This research received no external funding.

**Acknowledgments:** The authors of this work wish to gratefully acknowledge the Universitat Politècnica de València (UPV) through its research program, and the Institut Européen des Membranes (IEM). Special thanks are addressed to Salvador Cardona (UPV), Yves-Marie Legrand (IEM), and Eddy Petit (IEM) for their help in the research assistance.

**Conflicts of Interest:** The authors declare no conflict of interest.

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
