# Peer review of "Identification of Foulants on Polyethersulfone Membranes Used to Remove Colloids and Dissolved Matter from Paper Mill Treated Effluent"

_water, doi:10.3390/w12020365_

Round 1

Reviewer 1 Report

The introduction lacks a state of the art that allows us to know if exactly there are other studies where this has been analyzed and the characteristics of the water fed if it comes from real industrial waters, which would complete the work.

Author Response

Dear Reviewer,

According to your suggestions, additional information has been added in introduction and, materials and methods paragraphs.

Best Regards,

Fernanda

Reviewer 2 Report

Membrane fouling caused by paperboard mill treated effluent (PMTE) was investigated in this study, based on a Dead-end ultrafiltration (UF) pilot-scale study. Results of the study may have important application in the field of membrane filtration for wastewater treatment.  Authors may wish to consider the following in revision of their manuscript.

Please comment on the limitations of the proposed membrane filtration for treatment of paper mill treated effluent. Please include pre treatment used in treating paper mill wastewater prior to membrane filtration study. Please comment on effect of types of pre treatment processes prior to membrane filtration on the fouling of membrane in membrane filtration. Please include duration of membrane fouling study. Please explain why long term operation study was not used in the membrane fouling study. Please present data regarding what % of membrane fouling which is due to organic substances and what % of membrane fouling which is due to inorganic substances. Please present BOD and TOC data of treated effluent from membrane filtration. What are the effluent standards for paper mill wastewater treatment in author’s country. Please comment whether proposed treatment process could produce effluent which will meet effluent standards in author’s country.

Author Response

(The authors gave the same response as above.)

Reviewer 3 Report

This is a study that experimental results are predominantly shown. The reviewer does not question the extensive experimental research behind the project.However these have not been made relevant as to either innovation in membrane processing or significance to the fields. There is lack of novelty and explanation as to why this study is important. There are numerous published studies in membrane fouling regarding ultrafiltration, what does this study offer that is different to what is already known?

Author Response

(The authors gave the same response as above.)

Round 2

Reviewer 1 Report

Intruction IS poor.

Author Response

Dear Reviewer,

We welcome your comments about the introduction.   We have considered your last comment and we have rewritten the introduction so that the text can be better understood.   Best Regards, Fernanda